# Physiologically-Based Pharmacokinetics Modeling for Hydroxychloroquine as a Treatment for Malaria and Optimized Dosing Regimens for Different Populations

**DOI:** 10.3390/jpm12050796

**Published:** 2022-05-14

**Authors:** Jingchen Zhai, Beihong Ji, Lianjin Cai, Shuhan Liu, Yuchen Sun, Junmei Wang

**Affiliations:** Department of Pharmaceutical Sciences and Computational Chemical Genomics Screening Center, School of Pharmacy, University of Pittsburgh, Pittsburgh, PA 15261, USA; jiz183@pitt.edu (J.Z.); bej22@pitt.edu (B.J.); lic154@pitt.edu (L.C.); shuliu@med.umich.edu (S.L.); yus93@pitt.edu (Y.S.)

**Keywords:** hydroxychloroquine, PBPK modeling, SimCYP, ADMET, malaria

## Abstract

Malaria is a severe parasite infectious disease with high fatality. As one of the approved treatments of this disease, hydroxychloroquine (HCQ) lacks clinical administration guidelines for patients with special health conditions and co-morbidities. This may result in improper dosing for different populations and lead them to suffer from severe side effects. One of the most important toxicities of HCQ overdose is cardiotoxicity. In this study, we built and validated a physiologically based pharmacokinetic modeling (PBPK) model for HCQ. With the full-PBPK model, we predicted the pharmacokinetic (PK) profile for malaria patients without other co-morbidities under the HCQ dosing regimen suggested by Food and Drug Administration (FDA) guidance. The PK profiles for different special populations were also predicted and compared to the normal population. Moreover, we proposed a series of adjusted dosing regimens for different populations with special health conditions and predicted the concentration-time (C-T) curve of the drug plasma concentration in these populations which include the pregnant population, elderly population, RA patients, and renal impairment populations. The recommended special population-dependent dosage regimens can maintain the similar drug levels observed in the virtual healthy population under the original dosing regimen provided by FDA. Last, we developed mathematic formulas for predicting dosage based on a patient’s body measurements and two indexes of renal function (glomerular filtration rate and serum creatine level) for the pediatric and morbidly obese populations. Those formulas can facilitate personalized treatment of this disease. We hope to provide some advice to clinical practice when taking HCQ as a treatment for malaria patients with special health conditions or co-morbidities so that they will not suffer from severe side effects due to higher drug plasma concentration, especially cardiotoxicity.

## 1. Introduction

Malaria is one of the world’s most important parasitic infections, with high mortality rates in all ages, which mostly takes place in Africa and South-East Asia [1,2,3]. This disease results from repeated cycles of *Plasmodium* growth in erythrocyte [4], and clinical symptoms include fever, headache, chills, dizziness, diarrhea, vomiting, etc. [5].

Hydroxychloroquine (HCQ) is a Food and Drug Administration (FDA)-approved drug for malaria prevention and treatment [6]. According to the administration of FDA guidance, a starting dose of 620 mg HCQ base followed by 310 mg HCQ base at 6 h, 24 h, and 48 h can have treatment effect for malaria disease. Alternatively, the dosing regimen of 10 mg/kg base (not to exceed 660 mg base) followed by 5 mg/kg (not to exceed 310 mg base) at 6 h, 24 h, and 48 h provides guidance for malaria patients with lower body weight [6]. Chloroquine is also an approved drug that serves as a prevention and treatment drug for malaria [7]. However, compared to chloroquine, hydroxychloroquine has its advantages that need to be noticed. For example, it has less cardiotoxicity, it is more soluble, and has a higher tolerable dose [8,9,10,11,12,13].

Although this drug has been approved to be administered to the pregnant population, pediatric population, and other populations with special physical conditions, there is still no detailed guidance on the adjustment of dosages during clinical practice for these special populations. Besides, many patients with co-morbidities tend to have compromised renal function, which will lead the drug to more easily accumulate in the body. Thus, the original dosage provided by FDA guidelines may lead to more severe side effects caused by high drug plasma concentrations, including irreversible retinal damage, cardiomyopathy, QT prolongation, and other less frequently occuring consequences, such as worsening of psoriasis and porphyria, proximal myopathy and neuropathy, neuropsychiatric events, and hypoglycemia [14,15,16].

Physiologically based pharmacokinetic (PBPK) modeling is a bottom-up mathematical pharmacokinetic (PK) modeling method which can help predict the absorption, distribution, metabolism, excretion, and toxicity (ADMET) of drugs, based on available drug property information, dosing scenarios, and properties of virtual populations [17]. In this study, we used an in silico method to build the PBPK model of HCQ. We first predicted the PK profiles of HCQ under the original dosing regimen from FDA guidelines using the virtual healthy population and proposed a series of adjusted dosing regimens for several special populations with various physical conditions. The predicted concentration-time (C-T) curves for the adjusted dosing regimens for different populations are very close to the C-T curve for the healthy population under the original dosing suggested by FDA. The reduced dosing regimens advised by this research for different populations should have a similar treatment effect to the original dosing regimen provided by FDA for malaria patients without other diseases or special physical conditions, and can also circumvent severe side effects for these special populations [18,19].

## 2. Method

In this study, a PBPK model for HCQ was developed using SimCYP Simulator (V19, Release 1; Sheffield, UK) [20] software, which has a built-in drug database including information and PK templates of a series of commonly used drugs. All the simulations are based on the virtual population from SimCYP, and every simulation scenario includes 5 trials, with each trial including 10 subjects. Parameters of some enzyme activities are unavailable in clinical and preclinical reports, therefore, we used ADMET Predictor (V9.5, Simulation Plus, Lancaster, CA, USA) [21,22] to predict metabolic enzyme activity. Meanwhile, the advanced compartmental and transit (ACAT) model, which is utilized in the ADMET Predictor software to predict drug exposure [23], shares the most similarity with the advanced dissolution, absorption, and metabolism (ADAM) model adopted to analyze drug absorption in SimCYP. Thus, we chose the ADAM model to analyze drug distribution during the simulation. All the regression analysis were conducted using Microsoft Excel (Version 2112, Redmond, WA, USA) [24].

### 2.1. The PBPK Model Construction and Verification for HCQ

Recently, we have developed a computational protocol to develop high-quality PBPK models for arbitrary compounds which lack clinical pharmacokinetic experiment data [25]. To build up a PBPK model for a target drug, we identify a template for which its PBPK model is available based on Tanimoto structural similarity. Tanimoto score is an indicator that shows a fingerprint-based similarity between two compounds, which is calculated on the ChemMine Tools website (https://chemminetools.ucr.edu/similarity/, accessed on 15 November 2021) [26]. Before building the PBPK model for HCQ, we first calculated the Tanimoto score between HCQ and drugs (including substrates and inhibitors) in the SimCYP drug library. Among the limited drugs in the SimCYP database with well-built templates, Ciprofloxacin shares the highest Tanimoto score with HCQ (0.3056) and, thus, was selected as a template to construct the HCQ PBPK model. Except for drug properties collected from literature reports and predicted metabolizing parameters, absent or unclear information of HCQ, including blood-to-plasma partition ratio(B/P), the fraction of unbound drug in plasma (F_u_), and the p-gp (ABCB1) transporter were adjusted and compromised with each other within the range reported in the literature to fit the clinical reported PK curve of HCQ [27,28,29,30,31,32,33,34,35]. Details of input parameters are shown in Table 1.

### 2.2. The Inclusion of Different Populations

With the constructed HCQ model, we conducted a series of simulations to study drug plasma levels under different dosing regimens. Based on the FDA guidelines of HCQ administration to malaria patients [6], we extrapolated the safe and effective drug plasma concentration level for patients without any co-morbidities using the virtual healthy volunteers in SimCYP library. Accordingly, simulations were conducted under the following different virtual populations in SimCYP library and the C-T curves for them were predicted. Special populations include mild, moderate, and severe renal impairment (GRFL, GRFM, and GFRS) populations; morbidly obese (MO) population; geriatric Northern European Caucasians (NEC) population; obese population; pregnant population; pediatric; and rheumatoid arthritis (RA) populations. The renal impairment populations were taken into consideration because patients accepting HCQ treatment with renal impairment conditions have been reported in several studies to have retinal toxicity [36,37], hypoglycemia [38], and QT interval prolongation caused by delayed HCQ excretion during the treatment because of reduced renal function [39,40], while the administration of HCQ in patients with renal disease is still not well documented. MO population was included in this research because their body weight will lead to a lower drug plasma level [41], thus leads to reduced treatment effect. The geriatric and pediatric populations usually have special liver and renal function for drug disposition, indicating their special need for dosing adjustment [42,43]. The RA population was also taken into special consideration because HCQ is a drug approved to treat RA patients, and these patients have a special ADME response to HCQ [44].

## 3. Results

### 3.1. The PBPK Model for HCQ

The PK profiles predicted by the constructed model for HCQ are shown in Figure 1. The simulated T_Max_ under 200 mg HCQ sulfate is 3.46 h, compared with FDA guidance of 3.26 h and literature report ranging from 2–4.5 h, respectively [6,28,45]. Simulated C_Max_ reported a value of 207.41 ng/mL, while the literature report C_Max_ is 188–427 ng/mL when the same dose was applied [28]. The clinical time-concentration profile of HCQ data was also extracted from literature and overlayed in Figure 1 [28], with all the data points located within the confidence intervals. The good alignment of our model and clinical data validates the reliability of our PBPK model.

Multiple-dose validation has also been conducted on our PBPK model for HCQ (Figure 2). Compared to the clinical PK data of administrating HCQ 400 mg/week to 61 soldier patients who were infected with malaria [45], 29 out of 52 data points located in the predicted confidence interval from our model. More encouragingly, our model shows better performance compared to the previously published PBPK model for HCQ in this report [45].

### 3.2. Simulation on the Original Dosage Regimen for Different Special Populations

According to the FDA guidance, the treatment dosing regimen for malaria is 620 mg base followed by 310 mg base at 6 h, 24 h, and 48 h for adults. We first studied the drug plasma concentration of the virtual healthy population (Figure 3) and all different special populations (Figure 4) except the pediatric population under this dosing regimen. One reason of omitting pediatric population is that the age, height, and weight parameters vary a lot among the randomly generated virtual pediatric population. As shown in Figure 4**,** under the circumstance of the same dosing regimen, only the obese and MO populations show lower drug concentration levels than the healthy population, and all other special populations show higher drug levels in their simulated profiles. The pregnant population has the closest PK profile compared with healthy volunteers, and the RA patients are the next closest of their average C-T profile to the normal population. In contrast, the renal impairment populations with different severity levels and the elderly population show dramatic differences with the healthy volunteers.

### 3.3. Dosing Adjustment Recommendation for Different Populations

For the sake of comparison of the drug concentration levels after dosage adjustment for different special populations from that obtained by the administration of the original treatment dose in the healthy population, the four peak concentrations after four doses in healthy population with the original treatment regimen are defined as CP1 (815 ng/mL), CP2 (1010 ng/mL), CP3 (837 ng/mL), and CP4(860 ng/mL), respectively (Figure 3). Similarly, for the sake of simplicity during the PK profile comparison, we refer to the PK profile of the healthy population under the original dose as the reference PK profile or the reference C-T profile in the rest of the text.

#### 3.3.1. Pregnancy Population

The prediction result for the pregnancy population under the original dosing regimen was compared to the healthy volunteers with the same dose in Figure 5. The pregnancy population still possesses a C-T profile similar to the reference profile; thus, not much dosing adjustment is needed for this special population. Here, we proposed an adjusted dosing regimen, with the first dose reduced from 610 mg to 465 mg (1 tablet reduced) and other dosing unchanged (Reduced dose 1). As exhibited in Figure 5, the PK profile of the Reduced dosing 1 after the first dose is lower than the original dose in healthy population (CP1), but the drug concentration level after the second dose in this reduced regimen is very close to that for healthy volunteers under regular dosage (CP2). Furthermore, the C-T curve shows a higher similarity between the reduced dose for the pregnancy population and the original dose for the healthy population after the third dose given. In conclusion, the adjusted dosage for the pregnancy population gets the C-T profile in this population being closer to but not exceeding the highest blood level as seen in the healthy population.

#### 3.3.2. RA Population

The PK profile for the RA population under original dosing suggests the necessity to reduce dose because the second, third, and fourth peak drug concentrations are all higher than the maximum drug concentration occurring in the original treatment for healthy population (Figure 6). To investigate a proper dose for RA patients in which the drug PK profile can be close to the reference profile, we proposed four different reduced dosing regimens. Except for Reduced dose 1 for which one tablet is reduced only in the first dose, Reduced doses 2, 3, and 4 have one more tablet reduced in the second, third, and fourth dose, respectively. The detailed dosages for those modified dosing regimens are summarized in Table 2. The simulation results for the RA population under four different reduced dosing compared with the reference C-T profile are shown in Figure 6. It is demonstrated that Reduced dose 1 can lower the drug plasma concentration of the RA population and shows a similar C-T profile as the reference one, except that the drug concentration after the third and fourth doses is still slightly higher even though the two peak C_Max_ values are both below the maximum concentration of the reference profile (CP2). Based on this result, Reduced dose 2 was designed and its PK profile can better reproduce the reference profile with the first two peak concentrations (CP1 and CP2) lower than, and the last two (CP3 and CP4) closer to, the corresponding peaks of the reference profile. Similarly, Reduced dosing 3 and Reduced dosing 4 also decrease the drug peak concentration level at the time of the third and fourth dose, respectively. To summarize, reducing the first dose by one tablet is necessary to keep the drug concentration below CP2 in the whole treatment process, and whether the following doses should be reduced or not depends on the patients’ condition during clinical practice.

#### 3.3.3. NEC Population

Apparently, the predicted C-T for NEC population is above the reference C-T curve, and all the peak concentrations, except the first one, obviously exceed CP2 for the NEC population (Figure 7). To circumvent severe side effects caused by the unexpected high drug concentration, adjusting treatment dosing for this population is necessary. We first applied Reduced dose 1, although the peak concentrations are lower compared to the original dose, the peak concentrations except the first one also exceeded CP2 (Figure 7); as such, additional dose reduction is needed in the first three doses. Similarly, Reduced dose 2 also did not show a satisfactory prediction result because the drug peak concentration after the fourth dose is still much higher than CP3. Other two dosing regimens, Reduced dose 3 and 4, are unlikely to produce satisfactory PK profiles; thus, we designed Reduced dose 5, which further reduced the fourth dose by one tablet based on Reduced dose 2 (Table 2). Although the second drug peak concentration is lower than CP2, the drug level is very close to CP3, indicating that this drug level can reach a satisfactory treatment effect. Moreover, the first and third peak drug concentrations with this regimen are both very close to the those of the reference profile, and the fourth peak concentration in Reduced dose 5 is slightly lower than CP4. In addition, according to the prediction result of Reduced dose 1 in this population, the drug tends to accumulate more instead of keeping on a constant concentration range. Hence, we also proposed the Reduced dose 6, which reduced the dosing by one tablet at the first, third, and fourth doses (Table 2). The prediction result of this adjusted dosing regimen also shows an acceptable PK prediction result, with the overall shape and peak concentrations close to those of the reference profile. In summary, both Reduced 5 and 6 can be considered as suitable dosages for this special population.

#### 3.3.4. Renal Impairment Populations

Drug accumulation in renal impairment populations is much more severe, as illustrated in Figure 8. All four peak concentrations of drug in these renal impairment populations exceed CP2 under the original dose, even though the first peak concentrations are close to CP1. This demonstrates that the original dosing regimen can cause severe side effects. Reduced dose 1 on mild/moderate/severe renal impairment populations cannot lower the drug concentration level at the second, third, and fourth drug peak concentrations than CP2, as exhibited in Figure 8. The prediction C-T curves of Reduced dose 2 are lower than those of Reduced dose 1, but the fourth drug peak concentrations of all the renal impairment populations are still higher than CP2, suggesting additional dose reduction is needed to lower the drug concentration levels after the second dose. Reduced dose 3 and Reduced dose 4 were not tested as they are unlikely to have a better performance than Reduced dose 2. Thus, the Reduced dose 5 was conducted and its prediction results as shown in Figure 8 are satisfactory: the first drug peak concentrations are very close to the drug level at CP1 and significantly lower than CP2 for all renal impairment populations. Specifically, for the mild impairment population, the second drug peak concentration is between CP2 and CP3; the drug peak concentration after the third dose is slightly higher than CP3 but still lower than CP2; and the fourth drug peak concentration is lower than CP4 to a small extent. Similarly, for the moderate and severe renal impairment population, the first and the fourth drug peak concentrations are very close to the corresponding drug levels of CP1 and CP4; and the second and third drug peak concentrations are all around the corresponding drug concentration levels of CP2 and CP3. Furthermore, the peak concentrations in Reduced dose 5 for all the renal impairment populations are around the peak concentrations of the reference PK profile, ensuring the treatment effect under the reduced dosage. In short, Reduced dose 5 achieves the best performance in producing the reference PK profile and is therefore recommended for renal impairment populations.

#### 3.3.5. Obese Population and MO Population

The predicted PK profiles for the obese population and the MO population under the original dose are comparable to the healthy volunteers under the original treatment regimen as shown in Figure 9. The PK profiles for the obese population and the healthy volunteers are essentially similar, only the drug concentrations around the pick areas are slightly lower. In contrast, the predicted C-T curve for MO population is much more different from the reference C-T curve. Apparently, a dose increase is needed to reproduce the reference profile for this special population. 

However, considering that body weight is a very important factor that impacts drug concentration in MO patients and their actual body weight can vary a lot among individuals, we did not propose a specific increased dosing regimen for this special population. Instead, we conducted regression analysis to investigate the relationship between drug peak concentrations and patient physical condition for every generated virtual MO patient to achieve precision medicine. The involved physical parameters include body weight, height, glomerular filtration rate (GFR), and serum creatine level. The regression models include the exponential, linear, logarithmic, second-order polynomial, and power models. The coefficient of determination (R^2^ values) of all regression models for different parameters have been summarized in Table 3. Both body weight and height show slight correlations with drug plasma concentration, but the performances between different models are very close. As such, linear regression models are preferred due to its simplicity. The performance of linear regression models for weight and height are shown in Figure 10. In summary, we would recommend proportionally elevating the treatment dosing according to the patients’ body weight to achieve satisfactory treatment effect.

#### 3.3.6. Pediatric Population

Considering the pediatric population also have large interindividual variety, we also conducted regression analysis utilizing different type of models to analyze the correlation between interindividual drug peak concentrations and their body parameters. The model types are the same as when modeling the MO population, and the physical parameters investigated including body weight, height, age, and glomerular filtration rate (GFR). As demonstrated in Table 4, all the tested physical parameters have high correlation with the drug concentrations, and the largest R^2^ value in each row has/have been highlighted with bold text. For weight and GFR, the power model has the best performance with more top-ranked R square values, while the logarithmic model exhibits the best fitting performance for patient age and the exponential model displays the best performance for height. The selected regression model for each physical parameter along with the fitting equation and R^2^ value was shown in Figure 11.

In summary, the formulas shown in Figure 11 can provide guidance in clinical practice on adjusting dose to achieve effective treatment while minimize potential side effects for pediatric patients.

## 4. Discussion

The most commonly available dosage form of HCQ is 200 mg HCQ sulfate (equivalent to 155 mg base). In this study, all the mentioned dosing regimens, either the original one from FDA guidelines or the proposed dosing regimens for special populations, used a 155-mg-tablet as a dosing unit. The pregnant population, elderly population, and renal impairment populations all have different renal functions, which lead to different PK profiles after drug administration. Noticeably, it is reported that the ABCB1 transporter, which is a key transporter of HCQ, was overexpressed on RA patients [46]. This report indicates that RA patients are more likely to suffer from cardiotoxicity side effects than the elder population under the same drug plasma level. Although there are some dosing recommendations of HCQ for malaria treatment, the dosing adjustment guidance for patients with co-morbidities is still lacking, and the dosing reduction for pediatric and elder populations is not detailed enough for clinical application. Improper dosing, which contributes to a higher drug concentration than needed, may expose them to the risk of severe side effects. As the main cardiotoxicity of HCQ, QT prolongation can result in dangerous cases [11,47]. Thus, we predicted the PK profiles for this population under the original dosing regimen and compared their PK curves with the healthy virtual population to propose a series of adjusted dosing regimens. The reason we use the healthy virtual population instead of malaria virtual population is that SimCYP Simulator does not have a virtual population model for malaria population, and we do not have enough data to build one. This substitution should not significantly influence our research outcome and there is no study showing great difference of HCQ disposition between the malaria population and healthy population. Under an adjusted dosing regimen suggested for each population, the drug peak concentration levels are very close to the corresponding drug peak concentration levels under the original dosing for the healthy population. Furthermore, we guarantee that the maximum drug concentration for the recommended dosing regimen for a special population is not higher than CP2, the maximum drug concentration for the healthy population under the original dose. By this way, the recommended dosing regimens can avoid severe toxicity.

For the MO patients and pediatric population, we did not come up with specifically adjusted dosing regimens. The reason is that the interindividual varieties of the key physical parameters are very large. Instead, we conducted regression analysis for these populations and speculated the relationship between their physical condition and drug peak concentrations after different dosing. For the MO population, only weight and height show a slight correlation to drug concentration and different regression models have very close performance. Thus, it is more reasonable to choose simple linear regression models to guide dosing adjustment for an individual MO patient. However, due to the weak correlation between patient physical parameters and drug concentration level, it is still suggested that patients should be carefully monitored to avoid severe adverse effect when the treatment dosage is increased according to their physical parameters, especially the body weight. For the pediatric population, all the investigated physical parameters, including weight, height, and age, demonstrate high correlations with the drug concentration levels. Although the model type and the physical parameter of the best model for a specific peak concentration varies, the overall performance of the power models described by all physical parameters is satisfactory for all the four peak concentrations. It is worth pointing out that these physical parameters are also strongly correlated with each other for the pediatric population. The performance of regression models for each parameter does not mean that the change of this specific parameter alone can contribute to a great change in drug PK profile. However, this parameter can serve as a good indicator of the body maturity level for pediatric patients.

It is undeniable that the PBPK model can be further improved by collecting more experimental PK and clinical data as well as measured PBPK parameters. For example, there are two main transporters, SLCO1A2 and ABCB1, for HCQ [32,33]. However, only the parameters of ABCB1 can be modified in SimCYP Simulator software. In addition, the measured parameters for CYP enzymes are also not available and we relied on the ADMET Predictor module in the Simulations-Plus software package to predict CYP parameters. Thus, the prediction precision of ADMET is also a key factor that affects the quality of our PBPK model for HCQ. CYP 1A2 and 2D6 have been predicted as the major cytochrome P450 subtypes for HCQ by ADMET Predictor (Table 1). CYP 3A4 has also been reported to be involved in HCQ metabolism [48]; however, the parameters describing the metabolism reaction are still unknown. Thus, we did not include this less important metabolic pathway in our model for HCQ. For transporter parameters and blood binding parameters, we can only estimate the ranges for different parameters according to a variety of literature reports and adjust them to make the predicted C-T curve be overlayed with clinical reported C-T profiles to the most extent. Moreover, the reliability of our prediction inevitably depends on the performance of the SimCYP software itself. Nevertheless, we tried to build a PBPK model for HCQ to predict drug concentration under different dosing regimens. Encouragingly, the predicted C-T curves of our model can very well reproduce the measured plasma drug concentration during single-dose validation and our model has better prediction result than the previous HCQ model during multiple-dose validation. In addition, our model has better performance than the previously published PK model for HCQ with multiple-dose validation [45], reflected by more clinical data lying within the confidence interval of the predicted curve. In 2020, Yao et al. constructed a PBPK model using SimCYP Simulator for HCQ [49]; their model attracted some attention as it may be applied to optimize dosage regimen for treating COVID-19 patients [50,51]. We built a PBPK model using the reported parameters of their model; however, the prediction performance of the model is not as good as ours.

All the virtual populations involved in this study are all from the population models provided by SimCYP Simulator. Due to the complexity of co-morbidities and limited clinical research data, we cannot find enough clinical reports to validate our prediction for different special populations. However, this study provided some guidance to clinical application of HCQ. We revealed the differences between normal malaria patients and malaria patients with different physiological and physical conditions, and suggested dosage regimens of administrating HCQ to achieve effective treatment and minimize adverse effect simultaneously. However, carefully monitoring the patient condition is still highly recommended during the treatment of malarial using HCQ.

## 5. Conclusions

In this study, we successfully constructed a PBPK model for HCQ which helps to predict drug PK profiles based on clinical PK information. The original dosing regimen of HCQ to treat malaria suggested by FDA is for normal patients without the consideration of special physical conditions occurring in special populations. Based on this dosing guidance, we proposed a series of adjusted dosing regimens for several special populations. Reduced dose 1 is recommended to the pregnant population. Reduced dose 1, 2, 3, and 4 is applicable to RA patients according to their disease condition. The NEC population can consider Reduced dose 5 and 6. The Reduced dose 5 may also be considered by renal impairment populations. All the dosing details for the proposed reduced doses are provided in Table 2. We do not suggest changing dosing strategy for the obese and pediatric populations. The dosing level of MO population can be adjusted according to body weight or height. The pediatric population can adjust their dosing regimen with an eye towards the predicted drug concentrations using the established formulas described by patients’ physical parameters like height, weight and age. The current study may provide advice to clinical practice when taking HCQ as a treatment for malaria patients with special health conditions or co-morbidities so that they will not suffer from severe side effects due to higher drug plasma concentration, especially cardiotoxicity.

## Figures and Tables

**Figure 1 jpm-12-00796-f001:**
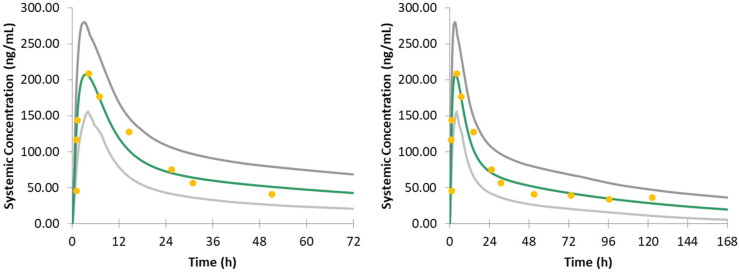
Validation of HCQ model within 3 days (**left**) and 7 days (**right**). The green curve represents the mean value of the simulated virtual subjects, and the grey curves represent the 90% confidence interval. The yellow dots represent the clinical reported 200 mg HCQ PK data.

**Figure 2 jpm-12-00796-f002:**
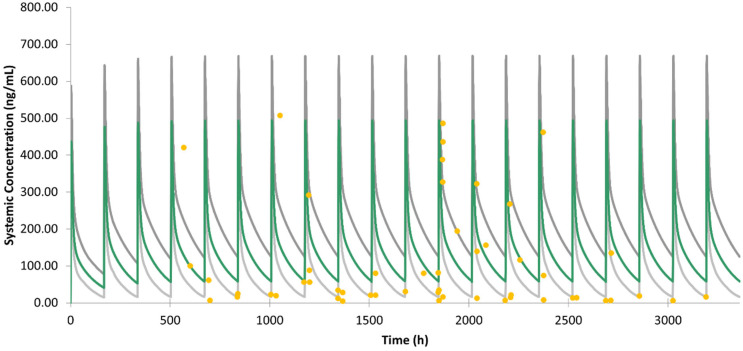
Validation of HCQ model under multiple dosing. The green curve represents the mean value of the simulated virtual subjects, and the grey curves represent the 90% confidence interval. The yellow dots represent the clinical reported PK data of 400 mg HCQ every week.

**Figure 3 jpm-12-00796-f003:**
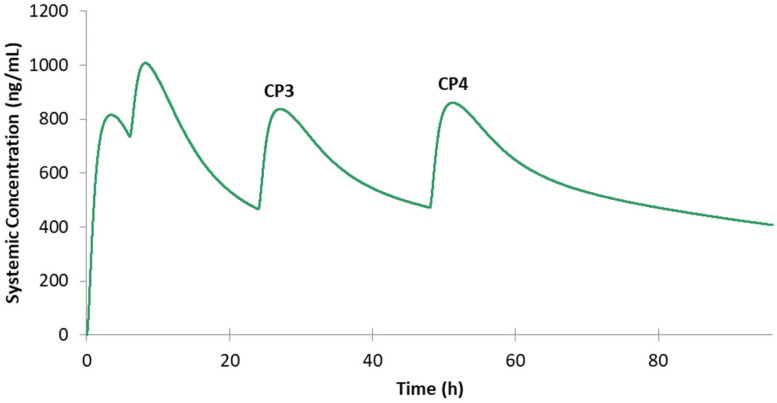
Predicted drug plasma concentration for healthy volunteers under the FDA-guided dosing regimen of 620 mg base followed by 310 mg base at 6 h, 24 h, and 48 h.

**Figure 4 jpm-12-00796-f004:**
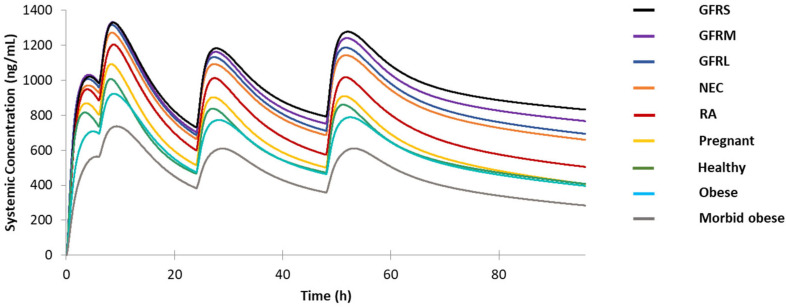
Predicted drug plasma concentration for healthy volunteers, pregnant population, NEC population, renal impairment populations, RA population obese population, and morbidly obese population under the same dosing regimen of 620 mg base followed by 310 mg base at 6 h, 24 h, and 48 h. GFRS: severe renal impairment populations, GFRM: moderate renal impairment population, GFRL: mild renal impairment population, NEC: geriatric Geriatric Northern European Caucasians population, RA: rheumatoid arthritis populations.

**Figure 5 jpm-12-00796-f005:**
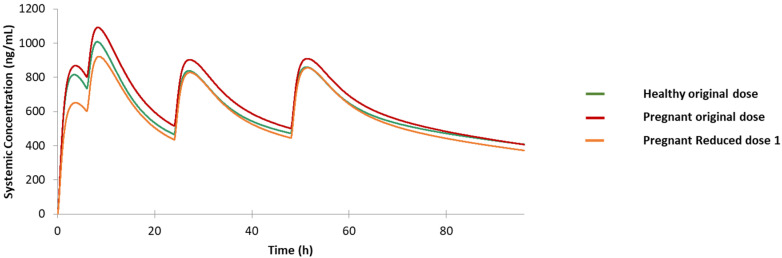
Predicted drug concentration for the healthy population under original dose, pregnancy population under original dose, and pregnancy population under Reduced dose 1.

**Figure 6 jpm-12-00796-f006:**
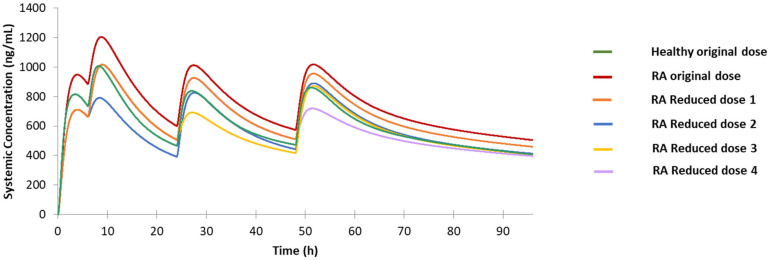
Predicted drug concentration for the healthy population under original dose, rheumatoid arthritis (RA) population under original dose, and RA population under Reduced dose 1, 2, 3, and 4.

**Figure 7 jpm-12-00796-f007:**
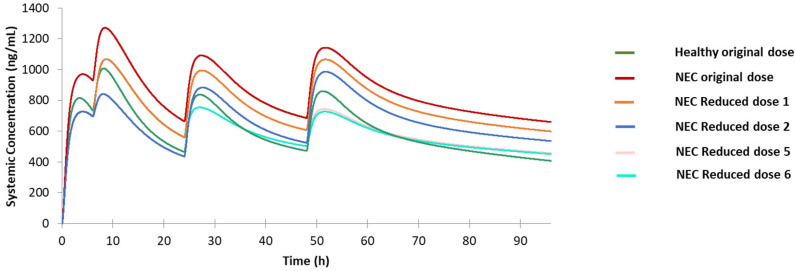
Predicted drug concentration for the healthy population under original dose, the geriatric Northern European Caucasians (NEC) population under original dose, and NEC population under Reduced dose 1, 2, 5, and 6.

**Figure 8 jpm-12-00796-f008:**
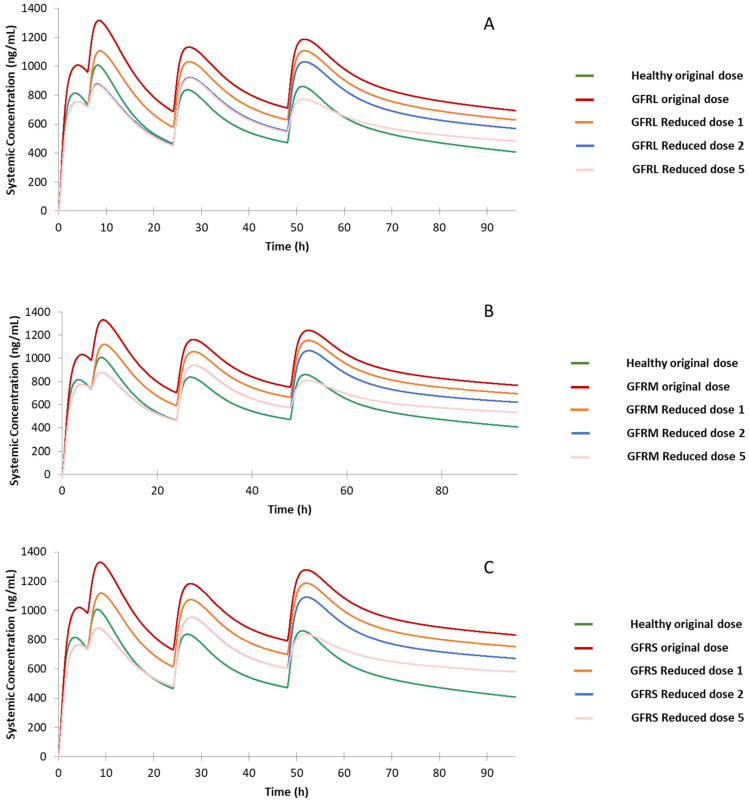
Predicted drug concentration for the healthy population under original dose, renal impairment populations under original dose, and renal impairment populations under Reduced dose 1, 2, and 5. (**A**): renal impairment mild. (**B**): renal impairment moderate. (**C**): renal impairment severe. GFRL: mild renal impairment population, GFRM: moderate renal impairment population, GFRS: severe renal impairment populations.

**Figure 9 jpm-12-00796-f009:**
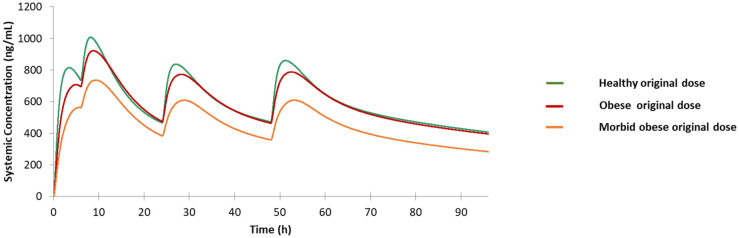
Predicted drug concentration for healthy population, obese population, and morbidly obese population under original dose.

**Figure 10 jpm-12-00796-f010:**
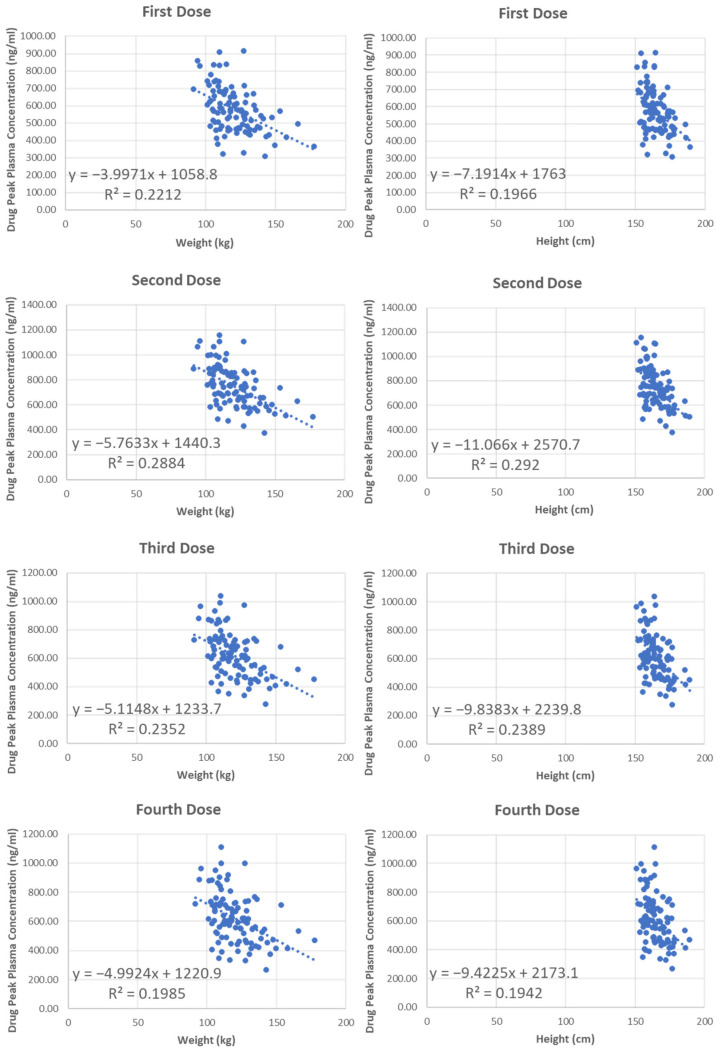
The linear regression models for MO population between individual physical parameters (weight and height) and drug peak concentrations after four doses. The *x*-axis represents the body weight, and the *y*-axis represents the peak plasma drug concentration after each dose.

**Figure 11 jpm-12-00796-f011:**
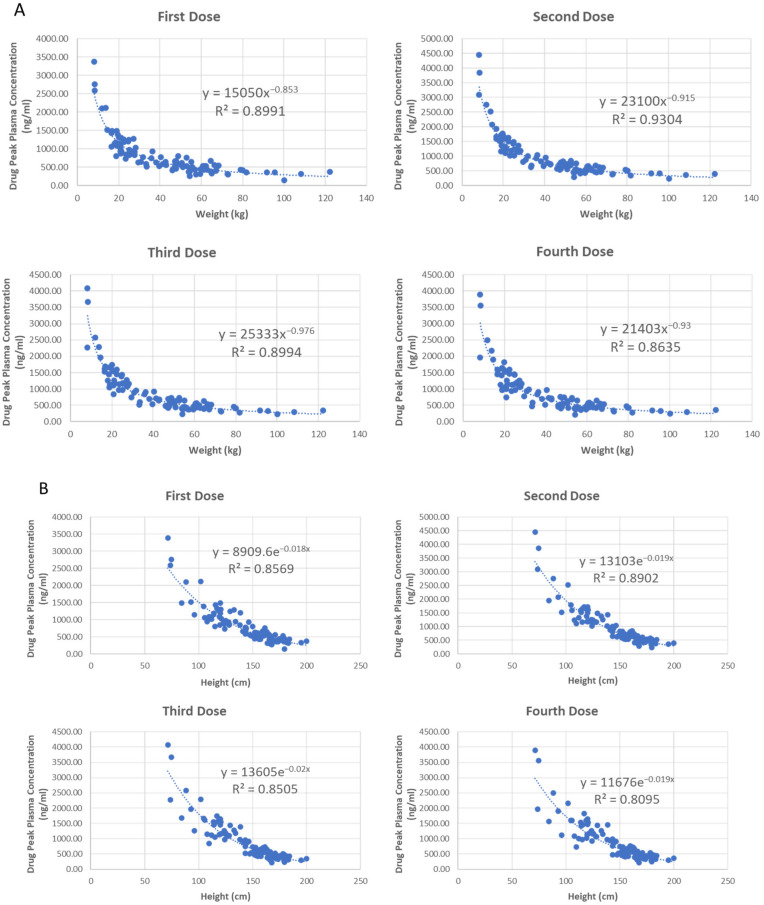
The regression models for pediatric population between physical parameters (*x*-axis: weight, height, age, and GFR) and drug peak concentrations after four doses (*y*-axis). (**A**): Power model. (**B**): Exponential model. (**C**): Logarithmic model (**D**): Power model.

**Table 1 jpm-12-00796-t001:** Input parameters for HCQ PBPK model. B/P: blood-to-plasma partition ratio. F_u_: the fraction of unbound drug in plasma. P_eff_: human jejunum effective permeability. PSA: polar surface area. V_ss_: volume of distribution at steady state using tissue volumes for a population representative of healthy volunteer population. Vmax: maximum rate of metabolism (pmol/min/pmol of isoform). Km: Michaelis-Menten constant, (μM). a: data from DrugBank. b: data from Pubmed. c: ADMET Predictor prediction result. d: fit clinical curve.

Parameter	Input Value
**Physiochemical Properties**	
Molecular weight (g/mol)	335.872 ^a^
LogP	3.6 ^b^
pKa	9.67 ^b^
**Blood Binding**	
B/P	0.55 ^d^
F_u_	0.1 ^d^
**Absorption (ADAM model)**	
P_eff_ (10^−4^ cm/s)	2.32 ^c^
PSA (*Å**^2^*)	48.39 ^a^
**Distribution (Full PBPK model)**	
Vss (L/kg)	SimCYP predicted
**Elimination**	
CYP1A2	Vmax: 7.928, Km: 20.777 ^c^
CYP2D6	Vmax: 2.319, Km: 14.602 ^c^
**Transporter**	
p-gp (ABCB1)	Clint: 18 ^d^

**Table 2 jpm-12-00796-t002:** Details of Reduced dose 1, 2, 3, 4, 5, and 6.

	Starting Dose	6 h	24 h	48 h
**Reduced dose 1**	465 mg	310 mg	310 mg	310 mg
**Reduced dose 2**	465 mg	155 mg	310 mg	310 mg
**Reduced dose 3**	465 mg	310 mg	155 mg	310 mg
**Reduced dose 4**	465 mg	310 mg	310 mg	155 mg
**Reduced dose 5**	465 mg	155 mg	310 mg	155 mg
**Reduced dose 6**	465 mg	310 mg	155 mg	155 mg

**Table 3 jpm-12-00796-t003:** The reported R^2^ value from the five different regression models for the parameters of the morbidly obesity (MO) population and the drug peak concentration after each dosing.

	**MO-First Dose**				
	**Exponential**	**Linear**	**Logarithmic**	**Polynomial**	**Power**
**Weight**	0.23	0.22	0.23	0.23	0.23
**GFR**	0.04	0.04	0.04	0.05	0.04
**Height**	0.19	0.20	0.20	0.20	0.19
**Serum creatine**	0.00	0.00	0.00	0.00	0.00
	**MO-Second dose**				
	**Exponential**	**Linear**	**Logarithmic**	**Polynomial**	**Power**
**Weight**	0.30	0.29	0.30	0.30	0.30
**GFR**	0.07	0.07	0.07	0.07	0.07
**Height**	0.29	0.29	0.29	0.29	0.29
**Serum creatine**	0.00	0.00	0.00	0.01	0.00
	**MO-Third dose**				
	**Exponential**	**Linear**	**Logarithmic**	**Polynomial**	**Power**
**Weight**	0.25	0.24	0.25	0.26	0.25
**GFR**	0.09	0.09	0.09	0.09	0.08
**Height**	0.24	0.24	0.24	0.24	0.24
**Serum creatine**	0.00	0.00	0.00	0.00	0.00
	**MO-Fourth dose**				
	**Exponential**	**Linear**	**Logarithmic**	**Polynomial**	**Power**
**Weight**	0.21	0.20	0.21	0.22	0.21
**GFR**	0.11	0.11	0.10	0.11	0.09
**Height**	0.19	0.19	0.19	0.19	0.19
**Serum creatine**	0.01	0.01	0.01	0.01	0.01

**Table 4 jpm-12-00796-t004:** The reported R^2^ value from the five different regression models for the parameters of the pediatric population and the drug peak concentration after each dosing.

	**Pediatric-First Dose**				
	**Exponential**	**Linear**	**Logarithmic**	**Polynomial**	**Power**
**Weight**	0.68	0.52	0.76	0.71	**0.90**
**Height**	0.86	0.75	0.82	0.85	**0.87**
**Age**	0.72	0.59	0.83	0.79	**0.85**
**GFR**	0.66	0.50	0.71	0.73	**0.79**
	**Pediatric-Second dose**				
	**Exponential**	**Linear**	**Logarithmic**	**Polynomial**	**Power**
**Weight**	0.71	0.53	0.78	0.73	**0.93**
**Height**	**0.89**	0.78	0.84	0.88	**0.89**
**Age**	0.76	0.62	**0.87**	0.82	**0.87**
**GFR**	0.70	0.53	0.74	0.76	**0.83**
	**Pediatric-Third dose**				
	**Exponential**	**Linear**	**Logarithmic**	**Polynomial**	**Power**
**Weight**	0.73	0.55	0.78	0.74	**0.90**
**Height**	**0.85**	0.77	0.82	0.84	0.83
**Age**	0.77	0.63	**0.84**	0.80	0.80
**GFR**	0.71	0.54	0.74	0.76	**0.79**
	**Pediatric-Fourth dose**				
	**Exponential**	**Linear**	**Logarithmic**	**Polynomial**	**Power**
**Weight**	0.71	0.54	0.76	0.72	**0.86**
**Height**	**0.81**	0.74	0.79	0.80	0.79
**Age**	0.73	0.61	**0.81**	0.76	0.77
**GFR**	0.69	0.53	0.72	0.74	**0.77**

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
