# Peer review of "Physiologically-Based Pharmacokinetics Modeling for Hydroxychloroquine as a Treatment for Malaria and Optimized Dosing Regimens for Different Populations"

_jpm, 2022, doi:10.3390/jpm12050796_

Round 1

Reviewer 1 Report

This manuscript, titled ‘Physiologically-Based Pharmacokinetics Modeling for Hydroxychloroquine as A Treatment for Malaria and Optimized Dosing Regimens for Different Populations’ by Zhai and et al., built a physiologically based pharmacokinetic modeling, called PBPK model, to provide hydroxychloroquine clinical administration guidelines for patients with special health conditions. Overall, the manuscript is easy to follow, and the study was designed and performed properly. I think the modeling and the reported dosage recommendations are teaching the field and providing some guidance to potential clinical dosage adjustment. Publication can be considered, but some revisions are necessary.

Authors concluded that ‘’The dosing level of MO population can be adjusted according to body weight or height’’ but given such a weak correlation with the body weight or height, this statement maybe not be supported. I guess my main question is that how do the authors justify that the selected patients’ physical parameters are suitable to predict the drug concentration levels? Especially for the MO population, where the problem lies in, tried any other predictors?

Some minor revisions:

  1. In the Abstract, change ‘comorbidities’ to ‘co-morbidities’ to keep consistency.
  2. Within section 2.2, for ‘pediatric and rheumatoid arthritis (RA) population’, add ‘s’ to ‘population’.
  3. Last, I suggest polishing a bit figure 10 and 11 to make them appear more professional.

Author Response

We sincerely thank the reviewer for their careful reading of this manuscript and the insightful suggestions and comments. Details of our response please find in the attached pdf file.

Reviewer 2 Report

The study of Zhai et al., provides a robust pharmacokinetic model for hydroxychloroquine, a key treatment of malaria to optimize the doses in different patients’ subpopulations; including pregnant ladies, Rheumatoid arthritis, renal impairment and morbid obese patients.

The study is of high importance in clinical practice and for clinical pharmacists recommending the appropriate dose of hydroxychloroquine.

-To emphasize the importance of the study, I suggest mentioning the adverse effects of hydroxychloroquine, briefly, in the introduction.

-Minor:

- Add all required abbreviations in the figure legends and table footnotes:

e.g., Figure 4 add all abbreviations, Table 3: MO for morbid obesity

- Revise grammar and spelling for the whole manuscript:

e.g., Page 15: 3.3.4 Renal impairment populations; line #7: does à dose

-language should be revised all through, e.g., in the “Abstract”, Malaria is infection disease; remove “disease”, or use “infectious disease”

The concluding statement should state “the current study provides/may provide advice to clinical practice when taking HCQ as a treatment for malaria patients with special health conditions or co-morbidities so that they will not suffer from severe side effects due to higher drug plasma concentration, especially cardiotoxicity.”

Author Response

(The authors gave the same response as above.)
